# Sustainable Dyeing and Functionalization of Different Fibers Using Orange Peel Extract’s Antioxidants

**DOI:** 10.3390/antiox11102059

**Published:** 2022-10-19

**Authors:** Aleksandra Ivanovska, Ivana Savić Gajić, Jelena Lađarević, Marija Milošević, Ivan Savić, Katarina Mihajlovski, Mirjana Kostić

**Affiliations:** 1Innovation Center of the Faculty of Technology and Metallurgy, University of Belgrade, Karnegijeva 4, 11000 Belgrade, Serbia; 2Faculty of Technology in Leskovac, University of Nis, Bulevar Oslobodjenja 124, 16000 Leskovac, Serbia; 3Faculty of Technology and Metallurgy, University of Belgrade, Karnegijeva 4, 11000 Belgrade, Serbia

**Keywords:** orange peel extract, sustainable dyeing, functionalization, wool, polyamide, cellulose acetate, antioxidant activity, antibacterial activity

## Abstract

A diluted ethanol orange peel extract was used for sustainable dyeing and functionalization of different fabrics. The extract analysis was performed using UPLC-ESI-MS/MS; its total flavonoid (0.67 g RE/100 g d.w.) and antioxidant (2.81 g GAE/100 g d.w.) contents and antioxidant activity (IC_50_ of 65.5 µg/mL) were also determined. The extract dyeing performance at various dyebath pH values was evaluated using multifiber fabric. Among six fabrics, extract possessed the ability for dyeing wool, polyamide, and cellulose acetate (at pH 4.5), which color strength (K/S) values increased after washing (9.7–19.8 vs. 11.6–23.2). Extract:water ratio of 20:35 (*v*/*v*) was found to be sufficient for achieving satisfactory K/S values (i.e., 20.17, 12.56, and 10.38 for wool, polyamide, and cellulose acetate, respectively) that were slightly changed after washing. The optimal dyeing temperatures for wool, polyamide, and cellulose acetate are 55, 35, and 25 °C, while the equilibrium dye exhaustion at those temperatures was achieved after 45, 120, and 90 min, respectively. The color coordinate measurements revealed that wool and polyamide fabrics are yellower than cellulose acetate, while, compared to polyamide and cellulose acetate, wool is redder. Possible interactions between selected fabrics and extract compounds are suggested. All fabrics possessed excellent antioxidant activity (88.6–99.6%) both before and after washing. Cellulose acetate provided maximum bacterial reduction (99.99%) for *Escherichia coli*, and *Staphylococcus aureus*, which in the case of *Staphylococcus aureus* remained unchanged after washing. Orange peel extract could be used for simultaneous dyeing and functionalization of wool and polyamide (excellent antioxidant activity) and cellulose acetate (excellent antioxidant and antibacterial activity) fabrics.

## 1. Introduction

The textile market is highly globalized wherein clothes are probably the most common items that people buy. Research from a McKinsey & Company (Chicago, IL, USA) [1] showed that the number of clothes produced annually has doubled since 2000 and exceeded 100 billion pieces in 2014. According to the European Environment Agency (EEA) [2], Europe’s clothing consumption (excluding leather and fur) in 2015, 2017, and 2019 accounted for 3.0, 3.3, and 3.5 million tons, respectively. These results are a clear sign that the demand for clothing is on an upward trend. Unfortunately, the majority of consumers are not aware that the textile industry is one of the world’s most polluting industries and has a huge impact on the Earth. Namely, water pollution caused by textile processing is very worrying, since, during the entire process of transformation from fiber to the final product, a great number of toxic chemicals are used. This especially applies to dyeing and finishing processes; it is estimated that about 20% of global water pollution is caused by dyeing and finishing textile products [3]. Based on The European Union’s 8th Environment Action Programme to 2030 (8th EAP; EU, 2020) [4] “pursuing a zeropollution ambition for a toxic-free environment, including for air, water and soil, and protecting the health and well-being of citizens from environment-related risks and impacts” is one of the six thematic priority objectives. Thus, there is a requirement for reducing the environmental and climate pressures from textile production and considering how circular business models can help move us toward a circular textile economy.

The researchers’ concern regarding the toxicity of textile industry effluents resulted in a wide variety of investigations focused on finding alternatives for textile dyeing, which include green chemistry principles, low waste generation, and recyclability/reusability of different agroindustrial wastes. For example, peels, shells, husks, roots, and wood-based extracts [5,6,7,8,9,10,11] have already been used as ecofriendly colorants for dyeing fibers of different chemical compositions. It is important to underline that some authors [12,13,14] have reported lower color strength (K/S) values and color fastness to washing of fibers dyed with natural colorants. To solve such drawbacks, metal-based salts such as CuSO_4_, FeSO_4_, ZnCl_2_, and KAl(SO_4_)_2_ have usually been used as mordants. Besides the fact that low amounts of these mordants have been used for improving fiber color strength, their toxicity makes the entire process unsustainable. Moreover, such mordants yield different colors with the same natural dye implying that the final color depends not solely on a dye but also on the type and concentration of the mordant that was used [15].

It is well known that dyeing conditions (pH, extract concentration, temperature, time) affect the fabric color fastness, color depth, and color hues, while the extraction conditions (technique, time, temperature, liquid-to-solid ratio, type of solvent) are responsible for the nature, quality, and content of extracted compounds. In [16,17], the authors have shown that the ultrasound-assisted extraction (UAE) of antioxidants from plum seeds and orange peels could be optimized using Box–Behnken design, central composite design, and numerical optimization method. According to the already conducted studies, orange peel extract is rich in polyphenolic compounds (scutellarein, catechin, rutin, quercetin, narirutin, hesperidin, nobiletin, tangeretin, quercetin-3-o-glucoside, including phenolic acids such as gallic acid, caffeic acid, etc.) [17,18,19,20,21,22,23]. Hesperidin, quercetin, and rutin, and their glycosides, are colored [24], while most of the previously mentioned components have antioxidant and/or antimicrobial activity [18,19,20,21,22]. Therefore, it is reasonable to assume that this extract has the potential to simultaneously dye and add a higher value to the textile. It is good to mention that Li et al. [25] have recently reported that tannic acid, which is found in the orange peel ethanol extract, has been used as a bio-based mordant during dyeing of cotton fabric. Some of the mentioned polyphenolic compounds found in the orange peel extract are ubiquitous, and relatively inexpensive, and could be used as commercially available chemicals for fabric dyeing. However, due to the depletion of natural resources, increasing greenhouse emissions, and awareness of the need for sustainable development in terms of safely reusing waste, the transformation of waste into valuable materials is emerging as a strong trend.

In light of the discussion above, this paper is focused on sustainable dyeing and functionalization of textiles of different chemical compositions using diluted ethanol orange peel extract in order to accomplish the concept “Less Waste, More Value”. The qualitative analysis of extract compounds was performed using UPLC-ESI-MS/MS. Furthermore, the orange peel extract was characterized in terms of its total flavonoid and antioxidant contents and antioxidant activity, while its dyeing performance was preliminarily evaluated on multifiber fabric at different dyebath pH values. Based on the obtained K/S values (after dyeing and washing), wool and cellulose acetate woven fabrics and polyamide knitted fabric were chosen for further investigations. After that, the extract concentration and dyeing temperature were optimized regarding the K/S values. Furthermore, dye exhaustion as a function of time was studied. Fabrics having the highest K/S values before and after washing were subjected to colorimetric measurements and their antioxidant and antibacterial activities were evaluated. Additionally, possible binding interactions between the extract compounds and fiber surface are considered.

## 2. Materials and Methods

### 2.1. Preparation and Characterization of Orange Peel Extract

The orange fruits were bought in the local market, and after the consumption of sweet oranges (*Citrus sinensis*), their peels were collected. Before the use as experimental material, the peels were dried at 50 °C for 48 h to the moisture content of 13.5% (*w*/*w*). The orange peel extract was prepared using ultrasound-assisted extraction according to the previously described procedure [17]. The extraction conditions were: 50% (*v*/*v*) ethanol, 30 min, 60 °C, and liquid-to-solid ratio of 15 mL/g. Before the analysis, the extract was filtered through a 0.45 μm filter.

The UPLC-MS/MS (Dionex Ultimate 3000 UHPLC+) equipped with a quaternary pump with a degasser, diode array detector (DAD), and LCQ Fleet Ion Trap Mass Spectrometer (Thermo Fisher Scientific, USA) was used for the qualitative analysis of the extract’s chemical composition. The instrument control, data acquisition, and data analysis were carried out using Xcalibur (version 2.2 SP1.48) and LCQ Fleet (version 2.1) software. Hypersil gold C18 column (50 mm × 2.1 mm, 1.9 μm) was used for chromatographic separation. The system was thermostatted at 25 °C, while the mobile phase having a flow rate of 0.25 mL/min consisted of phase A (0.1% formic acid in methanol) and phase B (0.1% formic acid in water). The linear gradient elution was applied for the separation of phenolic compounds in the regime as follows: 0–2 min (10–30% B), 2–4 min (30–35%), 4–5 min (35–40% B), 5–8 min (40–60%), and 8–11 min (60–90% B), followed by an isocratic period of 11–15 min (90% B), 15–15.01 min (90–10% B) and finished from 15.01 to 20 min by 10% B. The detection of bioactive compounds was carried out at the wavelengths of 280, 300, 350, and 590 nm. UV–Vis spectra were recorded in the full spectral range of 190–700 nm. The mass analysis was carried out using the 3D-ion trap with the electron spray ionization (ESI) in the negative and positive ionization modes. MS spectra were acquired in the full acquisition range of 100–900 m/z. The ESI conditions were as follows: source voltage of 4.5 kV, a capillary voltage of −41 V, tube lens voltage of −95 V, a capillary temperature of 350 °C, nitrogen sheath, and auxiliary gas flow of 32 and 8 arbitrary units, respectively. The collision-induced dissociation (CID) of detected molecular ion peaks ([M-H]^−^) was used for the fragmentation study. The normalized collision energy of the CID cell was adjusted at 30 eV. Based on the UV–Vis spectra of the DAD signals and the mass spectra, a qualitative analysis of the extract was obtained.

The total flavonoid expressed as gram rutin equivalent per 100 g of dry weight (g RE/100 g d.w.) was determined using the standard spectrophotometric procedure with aluminum chloride [26]. Folin–Ciocalteu’s reagent was used for the determination of total antioxidant content, which was expressed as gram gallic acid equivalent per 100 g of dry weight (g GAE/100 g d.w.). The 2,2-diphenyl-1-picrylhydrazyl assay was used to investigate the antioxidant potential of the extract according to the method given by Savic and Savic Gajic [17].

### 2.2. Utilization of Orange Peel Extract for Dyeing Fabrics of Different Chemical Compositions

The Multifiber Adjacent Fabric Style 42 (James Heal, England) was used for fast evaluation of the ability of prepared extract for dyeing fabrics of different chemical compositions such as worsted wool (WO), acrylic (PAN), polyester (PES), nylon (PA), bleached cotton (CO), and cellulose acetate (CA). One gram of multifiber fabric (4 cm × 10 cm) was dyed with 50 mL orange peel extract (extract:water ratio of 15:40 (*v*/*v*)) for 22 h with constant shaking in a Memmert water bath WNE 14 at 35 °C. The dyebath pH was varied from 2.5 up to 10.5 to investigate its influence on fabric K/S values. The multifiber fabrics were further washed with 5 g/L standard detergent at 40 °C for 30 min (ISO 105-C10 (2010)), rinsed with distilled water for 10 min, and dried at room temperature. Thereafter, K/S values of each component of multifiber fabric were evaluated and compared with the appropriate result obtained before washing.

Based on the performed screening experiments, dyeing of commercially produced wool and cellulose acetate woven fabrics, and polyamide 6.6 knitted fabric, was optimized in terms of extract concentration, dyeing temperature, and dye exhaustion time. Namely, WO, PA, and CA were dyed with different extract concentrations: 10, 15, 20, or 25 mL of extract were diluted in 45, 40, 35, or 30 mL of distilled water, respectively. The dyeing was performed at a solution pH of 4.5 for 22 h at 35 °C, fabric-to-liquid ratio of 1:50. In the next step of dyeing optimization, the dyeing temperature was varied from 25 °C up to 65 °C, while the solution pH, extract concentration, and dyeing duration remained constant (4.5, 20 mL extract diluted in 35 mL of distilled water, and 22 h). The dye exhaustion time was optimized under the following experimental conditions: solution pH of 4.5, 20 mL extract diluted in 35 mL of distilled water, and temperature of 25, 35, and 55 °C for CA, PA, and WO, respectively. The extract adsorption UV–Vis spectra were recorded (Shimadzu 1700) at time intervals of 15 min in the first hour and every 30 min up to 150 min.

### 2.3. Characterization of Dyed Fabrics

Kubelka–Munk equation [27] was used to calculate the fabric color strength (K/S) values based on the reflectances recorded on the UV–Vis 2600r (Shimadzu, Kyoto, Japan).

The colorimetric parameters (L, a*, b*) measured in the CIELab color space were determined using a Color i7 Benchtop Spectrophotometer (X-Rite, Grand Rapids, MI, USA) under illuminant D65 using the 10° standard observer.

The fabric antioxidant activity was determined by a 2,2′-azino-bis(3 ethylbenzthiazoline-6-sulfonic acid) (ABTS) assay following the method described by Glaser et al. [28].

The fabric antibacterial activity was tested against two microorganisms: *E. coli* ATCC 25922, and *S. aureus* ATCC 25923 according to the ASTM E 2149-01 (2001) standard.

## 3. Results and Discussion

### 3.1. Extract Characterization

In order to describe the origin of dye, antioxidant, and antibacterial activities of fabrics functionalized with orange peel extract, its chemical nature was assessed by using UPLC-MS/MS. The scanning was carried out in the positive and negative modes and the obtained results revealed the presence of nine phenolic compounds, Table 1.

From Table 1, it is evident that procyanidin-trimer was recorded in the positive mode; the molecular ion peak was noticed at *m*/*z* 867, while the fragmentation ions at *m*/*z* 579 and 289 originated from its dimer and monomer. Catechin gave the molecular ion peak at *m*/*z* 289 and fragmentation ion peak at *m*/*z* 245 after the loss of CO_2_. Moreover, cleavage of the A-ring of flavan-3-ol in its molecule caused the appearance of the fragmentation ion peak at *m*/*z* 205. The base peak at *m*/*z* 609 originated from rutin, and due to the loss of sugar moiety in its molecule, the fragment ion peak at *m*/*z* 300 appeared. After the fragmentation of the hesperidin molecule, the loss of sugar moiety led to the formation of fragment ion at *m*/*z* 301. Narirutin has a molecular ion peak at *m/z* 579, while the fragment ion at *m*/*z* 271 is due to the loss of O-diglycoside (*m*/*z* 308). The presence of quercetin-3-*O*-glucoside at the retention time of 8.54 min was confirmed based on the mass spectra with [M-H]^−^ at *m*/*z* 463. The quercetin molecular ion peak was noticed at *m*/*z* 301, while the mass fragmentation ions as the result of the retro-Diels–Alder reaction of [1,2A]^−^ and [1,2A-CO]^−^ were at *m*/*z* 179 and 151, respectively. It must be mentioned that nobiletin and tangeretin were recorded in the positive mode, Table 1. For nobiletin, a molecular ion peak was noticed at *m*/*z* 403, while the fragmentation ions at *m*/*z* 388, 373, and 342 appeared as a result of the loss of ^●^CH_3_, 2^●^CH_3_, and CO + H_2_O + ^●^CH_3_, respectively. The characteristic molecular ion peak of tangeretin occurred at *m*/*z* 373; its fragmentation ion peaks were obtained at *m*/*z* 358, 343, and 312 as a consequence of the loss of ^●^CH_3_, 2^●^CH_3_, and CO + H_2_O + ^●^CH_3_, respectively.

### 3.2. Using Orange Peel Extract for Dyeing Multifiber Fabric

The characterized orange peel ethanol extract was diluted in distilled water (15 mL of extract in 40 mL of water) and used as a natural colorant for fibers of different chemical compositions. Namely, Multifiber Adjacent Fabric Style 42 was used for fast screening the ability of extract for dyeing WO, PAN, PES, PA, CO, and CA. The dyeing temperature of 35 °C and duration of 22 h were fixed, while the dyebath pH was varied from 2.5 up to 10.5. This experimental variable was selected since it affects the interaction strength between the extract compounds and the functional groups on fiber surfaces and the stability of some extract compounds. The appearance of the dyed and washed multifiber fabrics along with the calculated fabric K/S values are given in Figure 1 and Table 2, respectively.

Among six studied fibers, orange peel extract has an affinity to dye WO, PA, and CA under the previously mentioned experimental conditions independently of the dyebath pH value, Figure 1. However, the fabric K/S values are pH-sensitive; the acidic conditions resulted in higher K/S values compared to the neutral and alkali conditions, Table 2.

Surprisingly, after washing, the WO and PA fabric appearance did not change (Figure 1) and their K/S values slightly increased (Table 2), which could be explained by the extract compound pH sensitivity as well as a higher pH value of the standard detergent used for washing. On the other hand, the K/S values of CA negligibly decreased after washing (Table 2), which could be attributed to the different binding mechanisms between the colored compounds of the orange peel extract and fabric. Namely, as the hydrophobic interactions between extract compounds and CA are dominant, it is clear that washing detergent changed the ionic strength of the solution, which contributed to lowering the intensity of such interactions.

Although somewhat higher K/S values were obtained when the dyeing was performed at pH 2.5, a dyebath pH of 4.5 was chosen for further experiments. This decision was made based on three facts: (1) the diluted orange peel extract has a pH of 4.5, enabling dyeing and functionalization without pH adjustment; (2) a high amount of strong acid should be added in order to adjust the dyebath pH to 2.5; and (3) dyeing at such low pH (2.5) can adversely affect the fabric properties.

### 3.3. Using Orange Peel Extract for Dyeing WO, PA, and CA Fabrics

Based on the preliminary experiments carried out on multifiber fabric, dyeing of wool and cellulose acetate woven fabrics and polyamide 6.6 knitted fabric was further optimized in terms of extract concentration and dyeing temperature, both regarding the K/S values before and after washing. Thereafter, dye exhaustion was monitored as a function of time and the fabric color coordinates were measured.

#### 3.3.1. Effect of Extract Concentration on Fabric K/S Values

In the first set of optimization experiments, the extract concentration was varied and the following combinations were chosen: 10, 15, 20, or 25 mL of extract were diluted in 45, 40, 35, or 30 mL of distilled water, respectively. During these experiments, temperature and dyeing duration were maintained constant, 35 °C and 22 h. The appearance of fabrics dyed with the lowest and highest extract concentrations is given in Figure 2a, while their K/S values are shown in Figure 2b.

The appearance of fabrics dyed with different orange peel extract concentrations (Figure 2a) undoubtedly indicated that as the extract concentration increased, the fabric shades became deeper, and hence, their K/S values increased (Figure 2b), which is in line with previously published data [33]. Besides the fact that dyeing was performed under the same conditions, WO, PA, and CA have different shades (Figure 2a), which could be explained by their different chemical compositions implying different interactions with the extract colored compounds (as discussed in detail below).

It seems that the third combination (20 mL of extract diluted in 35 mL of water) is the most appropriate for further optimization experiments since the differences between the K/S values of fabrics dyed with this and higher extract concentration (i.e., 25 mL of extract diluted in 30 mL of water) are below 5% (Figure 2b), while the volume of the used extract is higher in the second case. Independently on the used extract concentration, the order of fabrics concerning their K/S values is WO > PA > CA. It has to be emphasized that WO possesses very high K/S values ranging from 19.3 to 21.1, which is unusual for dyeing with a natural colorant without using a mordant. For comparison, Hou et al. [33] reported K/S values between 2 and 9 for WO dyed with different orange peel extract concentrations (without mordant) at three times higher temperature. These results confirm that extraction and dyeing conditions used in the current study were carefully chosen in order to obtain the highest fabric K/S values.

After washing, K/S values of WO, PA and CA (20 mL of extract diluted in 35 mL of water) decreased by 7.5, 12.8, and 22.9%, respectively, Figure 2b. A high reduction in K/S values, especially in the case of PA and CA could be explained by the relatively low temperature used for dyeing (35 °C). An attempt was made to improve this behavior by varying the dyeing temperature from 25 °C up to 65 °C.

#### 3.3.2. Effect of Dyeing Temperature on Fabric K/S Values

From the results shown in Figure 3, it is evident that there is no specific relation between the dyeing temperature and the WO fabric K/S values, which can be attributed to the partial degradation of extract compounds at higher temperatures. The situation is somewhat different in the case of PA; the fabric K/S values increase gradually as the dyeing temperature rises from 25 up to 55 °C, while further temperature increase results in lower K/S values. The variations between CA fabric K/S values are very low, ranging from 8.3–10.38, whereby the highest K/S value was obtained when the dyeing was carried out at 35 °C.

In general, WO, PA, and CA dyed at 55, 35, and 25 °C possess the lowest differences between the K/S values before and after washing. This is especially prominent in the case of WO fabric (Figure 3) due to the fiber nature and the higher dyeing temperature (55 °C), enabling fiber swelling and structure opening altogether resulting in easier entry of the orange peel color molecules into the fiber structure and their better binding [25].

#### 3.3.3. Time Required for Attaining Dye Exhaustion Equilibrium

From the economic and industrial point of view, the time required for achieving dye exhaustion equilibrium is an important factor in the whole fiber dyeing and functionalization process. The dye exhaustion dependence on time is presented in Figure 4. Only 45 min are required for achieving dye adsorption equilibrium in the case of WO fabric, while twice the time (90 min) is sufficient for dyeing CA. In the case of PA, dye adsorption equilibrium was attained after 120 min. As expected, the highest and the lowest dye exhaustions were observed for the WO and CA fabrics (21.5 vs. 9.56%, respectively).

The percentages of dye exhaustion are in line with the fabric color coordinates (L, a*, and b*) that are presented in Table 3. More precisely, WO having the highest percentage of adsorbed dye has the darkest hue (i.e., the lowest L value), while CA, characterized by the lowest percentage of dye adsorption, has the lightest hue (i.e., the highest L value). The same relationship between fabric lightness (i.e., L values) and dye exhaustion was previously observed for wool fabric dyed with mulberry wood extract [5]. Moreover, WO and PA are yellower (have higher b* values) than CA, and WO is redder (has a higher a* value) compared to PA and CA. After washing, fabric color coordinates changed only slightly; WO became redder and yellower, while PA and CA became greener and bluer.

All of the results presented above prove that, independent of fabric chemical composition, their K/S values before and after washing could be optimized by varying the dyebath pH, extract concentration, temperature, and time. The advantage of the chosen dyeing conditions can be clearly seen after the comparison of the current study results and previously published ones. For example, Hou et al. [33] dyed WO with orange peel extract at 70–100 °C without mordant and the reported K/S values are significantly lower (ranging between 4 and 10) than those presented in Figure 1. In other words, we attempted to reduce the dyeing temperature of WO (compared to the mentioned data), while maintaining high K/S values both before and after washing.

### 3.4. Mechanism of Extract Adsorption onto Studied Fabrics

In order to reveal the putative mechanism of extract adsorption, the nature and type of functional groups present in both orange peel extract compounds and WO, PA, and CA fibers should be taken into account. When considering the structure of the extract compounds (Figure 5), it is evident that narirutin, hesperidin, catechin, quercetin-3-*O*-glucoside, quercetin, rutin, and procyanidin-trimer bear numerous OH groups, and therefore, have high propensity to form hydrogen bonds. On the other hand, nobiletin and tangeretin have methoxy groups capable of forming hydrophobic interactions [34]. Compounds having glycosidic linkage in their structures (narirutin, hesperidin, quercetin-3-*O*-glucoside, and rutin) are larger and heavier than other extract compounds and exhibit a significant steric hindrance, which reduces their ability to diffuse through the dyebath and hence to establish effective interactions with the fiber surface [35].

Having in mind the diverse structures of the extract compounds, it could be stated that binding interactions with different fibers are complex and involve multiple interactions. Literature suggests that flavonoid-type compounds can establish different interactions with protein-like fibers such as ubiquitous van der Waals forces, hydrogen bonds, and electrostatic and hydrophobic interactions [36]. Previous studies have shown that the adsorption of quercetin onto PA fibers is significantly higher than the adsorption of rutin [35] and can establish more firm hydrophobic interactions with chitosan resins [37]. Figure 6 represents simplified binding interactions of quercetin and tangeretin, as examples, with WO, PA, and CA fiber surfaces, while other extract compounds are supposed to behave in a similar manner.

The adsorption of orange peel extract compounds on the WO and PA surfaces is highly affected by dyebath pH value. As seen in Figure 1 and Table 2, the uptake of the extract compounds onto the mentioned fibers declines in alkaline conditions, implying that stronger interactions are established in an acidic solution. Depending on their structures, flavonoid compounds present in the extract have pK*_a_* values between 6 and 10 [38,39,40], indicating that at chosen pH value (4.5) all compounds are dominantly in their neutral forms. Under acidic conditions, the binding of the extract compounds is mainly governed by hydrogen bonds between their OH groups and hydroxyl, amido, and carboxyl groups of the WO surface [33], Figure 6. Furthermore, as the wool possesses hydrophobic parts originating from nonpolar and aromatic amino acid residues, it is reasonable to assume that hydrophobic and π-π stacking (between aromatic moieties) interactions also participate in the overall binding mechanism [36,41], Figure 6. It should be noted that K/S values do not change notably by changing the solution pH value from 2.5 to 8.5 (21.22 vs. 17.43), suggesting a relatively good binding of the extract compounds at these pH values. The drastic decline of the WO fabric K/S value at pH 10.5 could be related to flavonoid pK*_a_* values, wherein the deprotonation of the molecules occurs causing significant repulsion with the negative WO surface (isoelectric point of 4.8 [33]) and hence poorer binding. PA fibers have amide groups orderly linked by alkyl chains while bearing a lower share of amino and carboxylic groups. As the extract compounds contain myriad OH groups, it can be suggested that interactions with PA proceed via hydrogen bonds. The isoelectric point of PA is estimated to be 5.63 [35], being positive below and negative above this value. Similar conclusions regarding the establishment of hydrogen bonds can be drawn as in the case of WO. In addition to the hydrogen bonding, owing to the amphiphilic nature of the PA fibers, hydrophobic interactions between fiber methylene groups and hydrocarbon moieties of the flavonoid compounds should be expected [35]. As reported by Tang et al. [41], hydrophobic interaction and van der Waals forces greatly contribute to the adsorption of natural polyphenols on PA fibers.

From Table 2, it is clear that pH value does not have a significant impact on the color strength of this fiber, suggesting that interactions unsusceptible to pH change are responsible for the binding of the extract compounds to CA. Considering the CA hydrophobic nature, it is evident that the interplay of C–H‧‧‧π interactions (between CA methyl groups and phenyl groups) [42,43] and hydrophobic interactions between alkyl groups play a major role in the binding mechanism, Figure 6. Certainly, it should be noted that hydrogen bonds are also involved in binding but to some minor extent.

### 3.5. Functional Properties of WO, PA, and CA Dyed with Orange Peel Extract

Having in mind the total antioxidant and flavonoid contents (2.81 g GAE/100 g d.w., and 0.67 g RE/100 g d.w., respectively) in the orange peel extract, its IC_50_ of 65.5 µg/mL, and the fact that oxidative stress along with the different microorganisms negatively affects people’s lives [44], WO, PA, and CA fabrics dyed at 55, 35 and 25 °C were evaluated for their multifunctionality. Namely, fabric antioxidant activity (Figure 7) was assessed according to the commonly used ABTS method, while their antibacterial activity was tested against two microbial strains, i.e., *E. coli*, and *S. aureus*, Table 4.

All dyed fabrics have excellent antioxidant activity (>96.5, Figure 7), which is expected since all of the UPLC-ESI-MS/MS biomolecules detected in the orange peel extract (Table 1) have powerful antioxidant and radical scavenging activities [18,19,20,21,22,45,46]. From the discussed binding mechanism (Figure 6), it can be observed that extract compounds are oriented at the fiber surface in such a way that numerous OH groups remain free and unconstrained. These groups are further able to participate in the reaction with ABTS radical cations and are considered responsible for the extraordinary antioxidant activities of WO, PA, and CA.

The durability of antioxidant activity towards washing was also studied and the results revealed that WO and CA antioxidant activities negligibly decreased after washing, while PA is characterized by around 10% lower antioxidant activity after washing, Figure 7. The orange peel extract dyed and functionalized fabrics can be considered for various applications such as functional clothing (WO as a surface fabric and CA as lining)—fine stockings and socks (PA) for people with sensitive skin or having irritation and inflammation of the skin, even pressure sores, since they are capable of reducing oxidative damage, which is the ultimate cause of major physiological skin disorders [47].

The multifunctionality of dyed fabric was further evaluated from the aspect of their antibacterial activities against *E. coli*, and *S. aureus*, Table 4. Interestingly, CA fabric provided maximal bacterial reduction against *E. coli*, and *S. aureus*, which in the case of *S. aureus* remained unchanged after washing. Unfortunately, WO and PA did not provide microbial reduction against these microorganisms. Different binding mechanisms of orange peel extract compounds with functional groups of fabrics of different chemical compositions along with the higher dyeing temperatures stand behind the absence of antibacterial activity in the case of WO and PA.

Therefore, it can be concluded that orange peel extract can be used in a one-step, simple, economical, and green strategy for sustainable dyeing, and for obtaining WO and PA fabrics with excellent antioxidant activity together with multifunctional CA fabrics having antioxidant and antibacterial activities. Moreover, the dried orange peel solid parts separated after the extraction could be further evaluated as adsorbents for different pollutants in aqueous solution and real water matrices, thus contributing to “closing the loop” of the orange peel lifecycle through recycling, resulting in environmental and economic benefits, which is in line with the Circular Economy Package (2022).

## 4. Conclusions

A diluted ethanol orange peel extract having total flavonoid and antioxidant contents of 0.67 g RE/100 g d.w. and 2.81 g GAE/100 g d.w., respectively, together with high antioxidant activity (IC_50_ of 65.5 µg/mL) was successfully used for sustainable dyeing and functionalization of wool, polyamide, and cellulose acetate. The optimal procedure implies dyeing with extract:water ratio of 20:35 (*v*/*v*) at a dyebath pH of 4.5. On the other hand, dyeing temperature and time required for attaining dye exhaustion equilibrium vary from among fibers. Namely, wool, polyamide, and cellulose acetate should be dyed at 55, 35, and 25 °C, for 45, 120, and 90 min, after which the equilibrium dye exhaustion is achieved. The last one is in line with the fabric appearance, i.e., lightness. It has to be emphasized that under the optimized dyeing conditions, wool, polyamide, and cellulose acetate possess very high K/S values (18.65, 12.56, and 9.38) that slightly decreased after washing. Color coordinate measurements indicate that wool and polyamide fabrics are yellower than cellulose acetate, while wool is redder than polyamide and cellulose acetate. Binding of the extract compounds with wool, polyamide, and cellulose acetate fiber surfaces proceeds via the interplay of different interactions, wherein, in the case of wool and polyamide, the dominant interactions with extract compounds are hydrogen bonds strengthened by hydrophobic and stacking interactions. Adsorption of extract compounds onto cellulose acetate fiber is, on the other hand, mainly governed by hydrophobic and C–H‧‧‧π interactions rendering different properties of cellulose acetate in comparison to wool and polyamide. All fabrics possess excellent antioxidant activity (88.6–99.6%) both before and after washing. Cellulose acetate provided maximum bacterial reduction (99.99%) for *E. coli*, and *S. aureus*, which in the case of *S. aureus* remained unchanged after washing. Finally, orange peel extract can be used in a one-step, simple, economical, and green strategy for sustainable dyeing and obtaining wool and polyamide fabrics with excellent antioxidant activity, and multifunctional cellulose acetate fabrics having antioxidant and antibacterial activities.

## Figures and Tables

**Figure 1 antioxidants-11-02059-f001:**
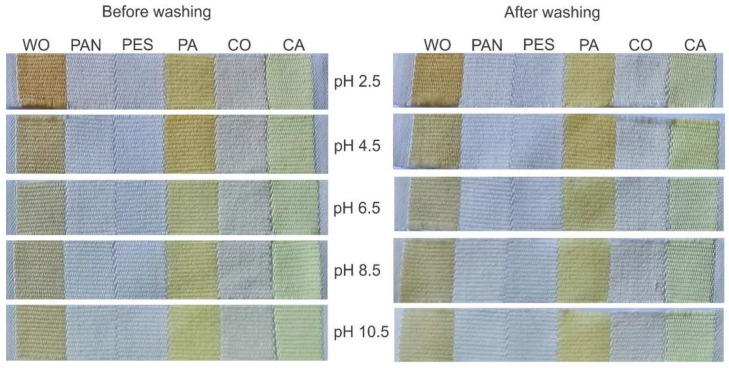
The appearance of multifiber fabric after dyeing with orange peel extract (15 mL extract diluted in 40 mL of water, 35 °C, 22 h) and washing.

**Figure 2 antioxidants-11-02059-f002:**
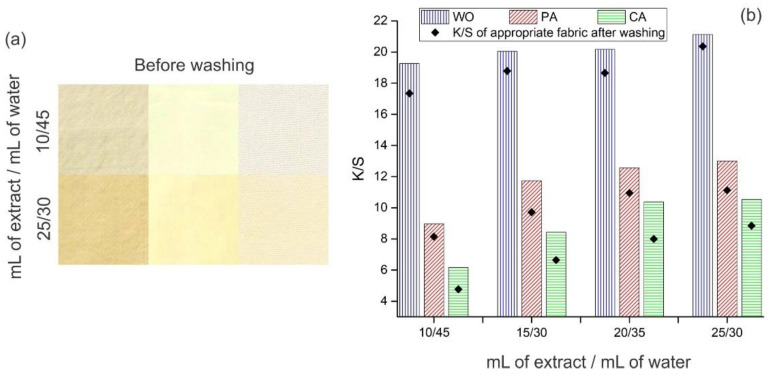
Fabric (**a**) appearance and (**b**) K/S values after dyeing with orange peel extract (pH 4.5, 35 °C, 22 h) and washing.

**Figure 3 antioxidants-11-02059-f003:**
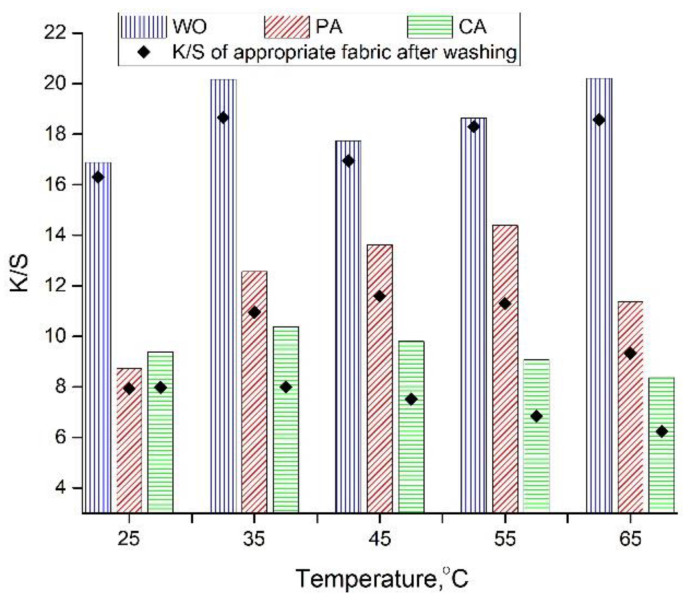
K/S values after dyeing with orange peel extract (pH 4.5, 20 mL of extract diluted in 35 mL of water, 22 h) and washing.

**Figure 4 antioxidants-11-02059-f004:**
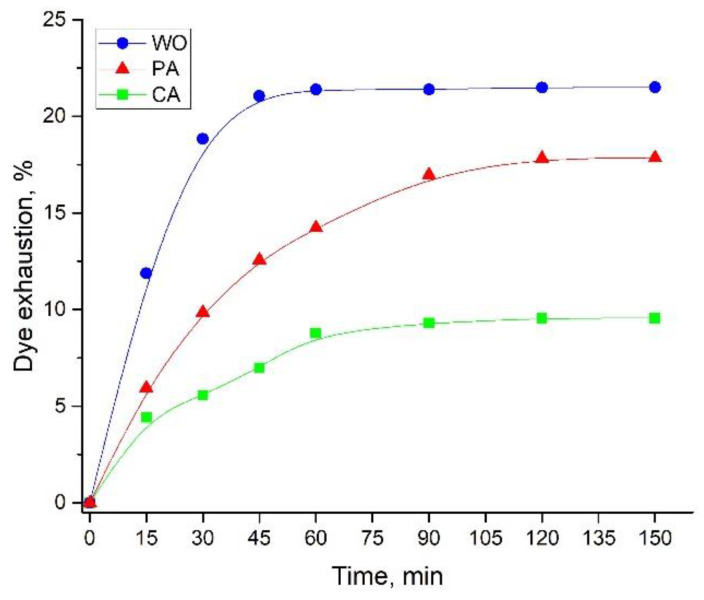
The influence of time on the dye exhaustion (pH 4.5, 20 mL of extract diluted in 35 mL of water, at 55, 35, and 25 °C for WO, PA, and CA, respectively).

**Figure 5 antioxidants-11-02059-f005:**
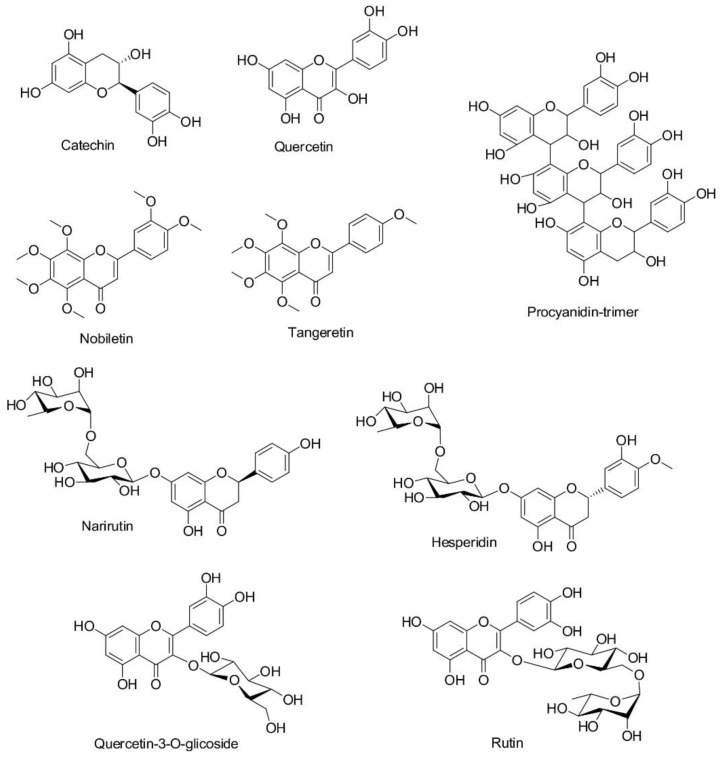
Molecular structures of the colored compounds present in the ethanol orange peel extract.

**Figure 6 antioxidants-11-02059-f006:**
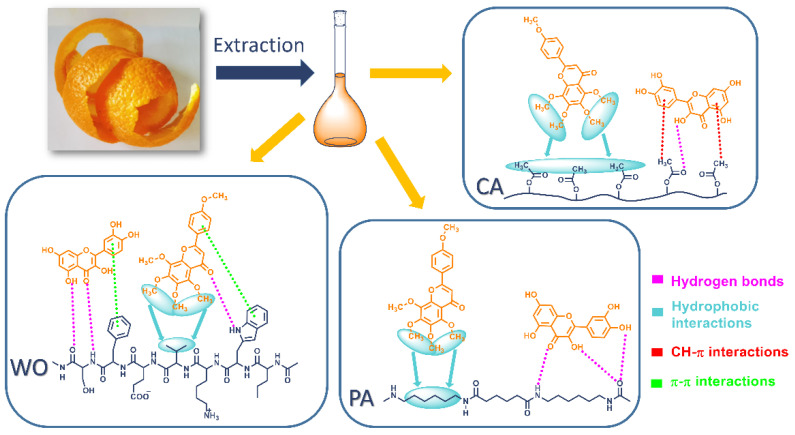
Schematic representation of the interactions between exemplar compounds (quercetin and tangeretin) and WO, PA, and CA fibers.

**Figure 7 antioxidants-11-02059-f007:**
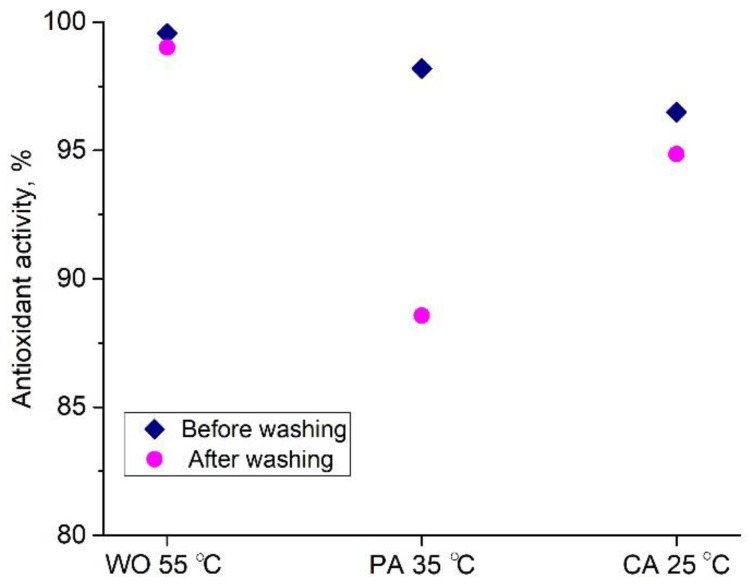
Antioxidant activity of selected dyed fabrics before and after washing.

**Table 1 antioxidants-11-02059-t001:** Chemical compounds in orange peel ethanol extract detected by UHPLC-ESI-MS/MS.

Compound	Retention Time, min	Molecular Ion Peak, *m/z*	MS/MS Fragment Ion Peaks, *m*/*z*	Mode	Reference
Procyanidin-trimer	5.96	867	579, 289	Positive	[29]
Catechin	6.12	289	245, 205	Negative	[30]
Rutin	8.74	609	300, 271	Negative	[30]
Hesperidin	8.78	609	301	Negative	[31]
Narirutin	8.88	579	271	Negative	[31]
Quercetin-3-*O*-glucoside	8.54	463	301	Negative	[30]
Quercetin	10.78	301	179, 151	Negative	[30]
Nobiletin	12.68	403	388, 373, 342	Positive	[32]
Tangeretin	13.90	373	358, 343, 312	Positive	[32]

**Table 2 antioxidants-11-02059-t002:** K/S values of multifiber fabrics presented in Figure 1.

Dyebath pH	Fabric	K/S before Washing	K/S after Washing
2.5	WO	21.22	23.00
	PA	10.71	13.44
	CA	18.13	17.73
4.5	WO	19.77	23.21
	PA	9.68	11.60
	CA	17.73	17.50
6.5	WO	18.08	19.47
	PA	5.86	6.81
	CA	17.07	16.84
8.5	WO	17.43	19.15
	PA	7.45	7.49
	CA	18.02	17.44
10.5	WO	8.16	9.24
	PA	7.13	6.74
	CA	18.15	17.71

**Table 3 antioxidants-11-02059-t003:** Fabric color coordinates before and after washing.

Fabric	Before Washing			After Washing		
	L	a*	b*	L	a*	b*
WO 55 °C	77.04	2.15	20.88	75.88	2.30	22.08
PA 35 °C	87.39	−0.97	21.95	89.17	−1.15	21.38
CA 25 °C	91.54	−1.91	14.80	93.43	−2.08	11.24

L—lightness or darkness; a*—the degree of redness (positive) and greenness (negative); and b*—the degree of yellowness (positive) and blueness (negative).

**Table 4 antioxidants-11-02059-t004:** Fabric antimicrobial activities before and after washing.

Fabric	Before Washing		After Washing	
	*Bacterial reduction, %*			
	*S. aureus*	*E. coli*	*S. aureus*	*E. coli*
WO 55 °C	/	/	/	/
PA 35 °C	/	/	/	/
CA 25 °C	99.99	99.99	99.99	/

/—no bacterial reduction.

## Data Availability

The data presented in this study are available on request from the corresponding author.

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
