# Peer review of "Sustainable Dyeing and Functionalization of Different Fibers Using Orange Peel Extract’s Antioxidants"

_antioxidants, 2022, doi:10.3390/antiox11102059_

Round 1
Reviewer 1 Report
Many natural dyes exhibit good bioactivities including antibacterial and antioxidant. Their antibacterial and antioxidant activities on textile fibers are important for medical care and health care. In this manuscript, authors extract natural dyes from orange peel and use them in textile dyeing and functionalization. The extracts are characterized by UPLC-ESI-MS/MS, the dyeing conditions as well as the colors of the dyed fabrics are studied, and finally antibacterial and antioxidant activities are tested. The results are positive. The research accords with the scope of ANTIOXIDANTS. The paper is clearly organized. The manuscript can be accepted for publishing after major revision.
Please note the following issues:
Abstract and Conclusions Section:
I suggest that authors should point out more important results, and had better not mention dyeing temperature and time.
Introduction Section:
Lines 64-66: “The utilization of the mordants could be bypassed by an appropriate selection of dyeing (pH, extract concentration, temperature, time) and extraction (used technique, time, temperature, liquid-to-solid ratio, type of solvent) conditions.”: I can not completely agree with authors at this point. It is well known that metallic mordants can improve the color fastness of naturally dyed fibers and broaden the hues of naturally dyed fibers. These factors that authors mention affect color fastness, color depth and color hues. But the appropriate selection of dyeing and extraction conditions can not substantively solve the problems of natural dyeing, in particular it can not solve the problem of low affinity of natural dyes to fibers, which leads to poor color fastness. So I suggest authors to improve the relative descriptions. Of course, the research on the affinity of different components extracted using different solvents to fibers is of great significance.
Lines 90-97: The detailed results or conclusions should not be written in the Introduction Section.
Materials and Methods Section:
In my opinion, the temperatures used for all fabrics are too low although long time is used. In practical production, wool and polyamide fabrics are dyed with acid dyes at 95-100 ℃, and cellulose acetate fabric is dyed with disperse dyes at 80-100 ℃, for the achievement of good color fastness due to good diffusion of dyed into fiber interior. I strongly suggest authors to use high temperature dyeing in the future work.
Results and Discussion Section:
Lines 264-271: Authors attempt to compare the color depth of their dyeing fabrics with that in previous reports. But some previous reports used other plant extracts and metallic mordants. In such cases, the comparison is not appropriate or not correct. These results can not be compared. I strongly suggest to authors to delete some of them. The same extracts with no use of mordants should be compared.
Lines 320-326: Authors attempt to compare the dyeing temperatures that they used with those in previous reports. But in my opinion, the temperatures used for all fabrics are too low although long time is used. Thus, such comparisons are of no significance.
I strongly suggest to authors to delete the comparison.
Lines 374-377: “On the other hand, taking into account the amphiphilic nature of the PA fibers it could be supposed that hydrogen bonds are additionally strengthened by hydrophobic interactions [42] between fiber alkyl chains and hydrocarbon parts of the flavonoid compounds [35]”: The description that hydrogen bonds are additionally strengthened by hydrophobic interactions is not appropriate. In Reference 42, the description that hydrogen bonds are additionally strengthened by hydrophobic interactions is not expressed, but the view that hydrophobic interactions greatly contribute to the adsorption of natural polyphenols on polyamide fibers is expressed. So I suggest authors to change the description in Lines 374-377.
Regarding antibacterial test results, wool and polyamide show great difference from cellulose acetate. I suggest authors to consider the antibacterial test principle (e.g., desorption or no desorption of antibacterial agent), and adsorption or no adsorption or low adsorption of antibacterial components in extracts on fibers.
Others:
I suggest authors to place the supplementary material into normal manuscript because dye structures are important.
English tenses should be improved.
By the way, I think that I have good background of textile dyeing. So I suggest authors to consider my suggestions about dyeing.
Reviewer 2 Report
1. As an important study material please provide more specific details on the orange peel - origin. Seasonal variations in phenolics?
2. Please comment - since a number of these phenolics are ubiquitous and several relatively inexpensive - in the long term in terms of product consistency wouldn't it be better to use the chemical available purified important dye phenolic(s) as a dye.
Author Response
Manuscript ID: antioxidants-1884103
Title: “Sustainable dyeing and functionalization of different fibers using orange peel extract’s antioxidants”
Respected Editor,
We would like to express our gratitude to the Reviewers for the given suggestions which shall improve the quality of our work. According to the Reviewers’ suggestions, we have carefully revised our manuscript. All changes and corrections are marked up using the “Track Changes” function.
We hope that you will consider the revised manuscript for publishing in Antioxidants.
Best regards,
Aleksandra Ivanovska
Reviewer #2:
Comment: As an important study material please provide more specific details on the orange peel - origin. Seasonal variations in phenolics?
Answer: We thank the Reviewer for raising this question. More specific details regarding the orange peels used in this experiment are added in section 2.1.
The orange fruits were bought in the local market, and after the consumption of sweet oranges (Citrus sinensis), their peels were collected. Before the use as experimental material, the peels were dried at 50 °C for 48 h to the moisture content of 13.5% (w/w).
Regarding the second part of the question, Hunluna et al. (2017) reported that the total antioxidant content did not differ significantly (p>0.05) with the seasonal variations.
Reference:
Hunlun, C., De Beer, D., Sigge, G. O., and Van Wyk, J. (2017). Characterisation of the flavonoid composition and total antioxidant capacity of juice from different citrus varieties from the Western Cape region. Journal of Food Composition and Analysis, 62, 115-125. https://doi.org/10.1016/j.jfca.2017.04.018
Comment: Please comment - since a number of these phenolics are ubiquitous and several relatively inexpensive - in the long term in terms of product consistency wouldn't it be better to use the chemical available purified important dye phenolic(s) as a dye.
Answer: We agree with the Reviewer that some of the phenolic compounds found in the orange peel extract are ubiquitous, and relatively inexpensive, and could be used as commercially available chemicals for fabric dyeing. However, due to the depletion of natural resources, increasing greenhouse emissions, and awareness of the need for sustainable development in terms of safely reusing waste, the transformation of waste into valuable materials is emerging as a strong trend. The reason stands behind the utilization of orange peel extract as a sustainable source of valuable compounds is to accomplish the concept “LESS WASTE, MORE VALUE”. Moreover, the dried orange peel solid parts separated after the extraction could be further evaluated as adsorbents for different pollutants in aqueous solution and real waters, thus contributing to “closing the loop” of the orange peel lifecycle through recycling resulting in environmental and economic benefits, which is in line with the Circular Economy Package (2022).
The following corrections were made in the introduction part of the manuscript:
According to the already conducted studies, orange peel extract is rich in polyphenolic compounds (scutellarein, catechin, rutin, quercetin, narirutin, hesperidin, nobiletin, tangeretin, quercetin-3-o-glucoside, including phenolic acids such as gallic acid, caffeic acid, etc.) [17–23]. Hesperidin, quercetin, and rutin, as well as their glycosides, are colored [24], while most of the previously mentioned components have antioxidant and/or antimicrobial activity [18–22]. So, it is reasonable to assume that this extract has the potential to simultaneously dye and add a higher value to the textile. It is good to mention that Li et al. [25] have recently reported that tannic acid, which is found in the orange peel ethanol extract has been used as a bio-based mordant during dyeing of cotton fabric. Some of the mentioned polyphenolic compounds found in the orange peel extract are ubiquitous, and relatively inexpensive, and could be used as commercially available chemicals for fabric dyeing. However, due to the depletion of natural resources, increasing greenhouse emissions, and awareness of the need for sustainable development in terms of safely reusing waste, the transformation of waste into valuable materials is emerging as a strong trend.
Additionally, the last paragraph in section 3.5. was rewritten as follows:
Namely, it can be concluded that orange peel extract could be used in a one-step, simple, economical, and green strategy for sustainable dyeing and obtaining WO and PA fabrics with excellent antioxidant activity as well as multifunctional CA fabrics having antioxidant and antibacterial activities. Moreover, the dried orange peel solid parts separated after the extraction could be further evaluated as adsorbents for different pollutants in aqueous solution and real waters, thus contributing to “closing the loop” of the orange peel lifecycle through recycling resulting in environmental and economic benefits, which is in line with the Circular Economy Package (2022).

Reviewer 3 Report
The present manuscript “Sustainable dyeing and functionalization of different fibers using orange peel extract’s antioxidants” studied the optimal dyeing conditions (pH, temperature, time, etc. ) for multifiber fabric using orange peel extract and followed by the functional properties (durability of antioxidant activity, antibacterial activities) of dyed WO, PA, and CA.
Overall, the manuscript is well-organized, the experimental is performed scientifically, the discussions and conclusions are logical and rational. Therefore, I would like to recommend it to Antioxidants.
Author Response
Manuscript ID: antioxidants-1884103
Title: “Sustainable dyeing and functionalization of different fibers using orange peel extract’s antioxidants”
Respected Editor,
We would like to express our gratitude to the Reviewers for the given suggestions which shall improve the quality of our work. According to the Reviewers’ suggestions, we have carefully revised our manuscript. All changes and corrections are marked up using the “Track Changes” function.
We hope that you will consider the revised manuscript for publishing in Antioxidants.
Best regards,
Aleksandra Ivanovska
Reviewer #3:
Comment: The present manuscript “Sustainable dyeing and functionalization of different fibers using orange peel extract’s antioxidants” studied the optimal dyeing conditions (pH, temperature, time, etc.) for multifiber fabric using orange peel extract and followed by the functional properties (durability of antioxidant activity, antibacterial activities) of dyed WO, PA, and CA. Overall, the manuscript is well-organized, the experimental is performed scientifically, the discussions and conclusions are logical and rational. Therefore, I would like to recommend it to Antioxidants.
Answer: We thank the Reviewer for the praises as well as the recommended acceptance of the manuscript.

Round 2
Reviewer 1 Report
This manuscript has been substantively improved. The present version can be accepted for publishing.
Author Response
Manuscript ID: antioxidants-1884103
Title: “Sustainable dyeing and functionalization of different fibers using orange peel extract’s antioxidants”
Respected Editor,
We would like to express our gratitude to the Reviewers for the given suggestions which shall improve the quality of our work. According to the Reviewers’ suggestions, we have carefully revised our manuscript. All changes and corrections are marked up using the “Track Changes” function.
We hope that you will consider the revised manuscript for publishing in Antioxidants.
Best regards,
Aleksandra Ivanovska
Reviewer #1:
Comment: This manuscript has been substantively improved. The present version can be accepted for publishing.
Answer: We thank the Reviewer for the praises as well as the recommended acceptance of the manuscript.